# The Amniotic Fluid Cell-Free Transcriptome Provides Novel Information about Fetal Development and Placental Cellular Dynamics

**DOI:** 10.3390/ijms22052612

**Published:** 2021-03-05

**Authors:** Hee Jin Park, Hee Young Cho, Dong Hyun Cha

**Affiliations:** Department of Obstetrics and Gynecology, CHA Gangnam Medical Center, CHA University, Seoul 06135, Korea; coolsome72@chamc.co.kr (H.J.P.); hycho.md@cha.ac.kr (H.Y.C.)

**Keywords:** amniotic fluid, placenta, cell-free mRNA, cell-free mRNA, cell-free transcriptome, genetic syndrome, pregnancy complication, fetal biomarkers, fetal development

## Abstract

The amniotic fluid (AF) is a complex biofluid that reflects fetal well-being during development. AF con be divided into two fractions, the supernatant and amniocytes. The supernatant contains cell-free components, including placenta-derived microparticles, protein, cell-free fetal DNA, and cell-free fetal RNA from the fetus. Cell-free mRNA (cfRNA) analysis holds a special position among high-throughput analyses, such as transcriptomics, proteomics, and metabolomics, owing to its ease of profiling. The AF cell-free transcriptome differs from the amniocyte transcriptome and alters with the progression of pregnancy and is often associated with the development of various organ systems including the fetal lung, skin, brain, pancreas, adrenal gland, gastrointestinal system, etc. The AF cell-free transcriptome is affected not only by normal physiologies, such as fetal sex, gestational age, and fetal maturity, but also by pathologic mechanisms such as maternal obesity, and genetic syndromes (Down, Edward, Turner, etc.), as well as pregnancy complications (preeclampsia, intrauterine growth restriction, preterm birth, etc.). cfRNA in the amniotic fluid originates from the placenta and fetal organs directly contacting the amniotic fluid as well as from the fetal plasma across the placenta. The AF transcriptome may reflect the fetal and placental development and therefore aid in the monitoring of normal and abnormal development.

## 1. Introduction

Cell-free fetal (cff) DNA analysis has been established for the non-invasive prenatal diagnosis of fetal sex, rhesus D status, and screening of fetal aneuploidies and micro-deletions [1,2,3]. In 1997, Lo et al. [4] identified fetus-derived Y sequences in the maternal circulation of women bearing male fetuses. This was the first report of cff DNA detection in maternal blood and served as the platform for the development of non-invasive prenatal diagnosis strategies currently in use. Subsequently, the discovery of uniquely methylated cff DNA in 2002 represented the first use of fetal epigenetic markers in the noninvasive prenatal analysis regardless of the fetal sex [5]. Since then, several studies have established that cff DNA originates from the placenta [6,7,8,9]. The dying cells (necrotic or apoptotic cells) are presumed to be the most abundant source of circulating DNA. Using electron microscopy, Baylor and colleagues [10] identified the presence of apoptotic bodies containing nucleosomes and chromatin in the plasma of pregnant women; this finding substantiated the hypothesis that fetal DNA may exist in distinct forms and fetal DNA enrichment is feasible. During pregnancy, the cff DNA is primarily placental in origin [11], and in normal pregnancies, grams of placental material are shed daily into the maternal circulation without causing inflammation [12]. This may be released from the syncytiotrophoblast layer of the placenta as a result of cellular apoptosis and necrosis [13]. For the past decade, circulating cff DNA has been investigated as a potential marker to predict complicated pregnancies [14,15,16,17,18]. Increased release of cff DNA into the maternal circulation is attributed to increased trophoblast apoptosis and shedding of syncytiotrophoblast microparticles due to increased oxidative stress and inflammatory response. Therefore, the placenta affected by ischemia in preeclampsia can increase cff DNA levels in maternal circulation [12,19]. The cff DNA also has the potential to induce pro-inflammatory signals, resulting in the systemic activation of maternal inflammation. Increased release of cff DNA into the maternal circulation may result in the activation of an inflammatory cascade that initiates preterm labor [20]. However, in clinical studies, the evidence linking cff DNA to adverse pregnancy outcomes including preterm birth and preeclampsia is inconsistent [14,18,21,22,23,24].

With the rapid development of non-invasive methods for fetal testing, maternal plasma has been subjected to focused research. However, as the placenta (not fetus) is a limited source of cff DNA, maternal blood contains mainly cell-free nucleic acids derived from maternal leukocytes with a limited fetal contribution of only 19% [25]. It is completely unexpected that there is circulating fetal mRNA in maternal blood because there are ribonucleases in maternal plasma and RNA is well known to be unstable in tissue. A previous study showed that fetal *ZFY* (zinc-finger protein) mRNA sequences were detected in the plasma of pregnant women expecting a male fetus [26]. mRNA in maternal blood is considerably stable because mRNA circulates within apoptotic bodies and is protected from degradation. In 2010, Bianchi et al. [27] demonstrated that fetal mRNA in maternal blood can indicate defects in fetal functional development, even if no structural abnormalities are detected via ultrasound In this study, 27 of the 157 differentially-regulated genes related to fetal development and 48% of those were associated with the development of the nervous system. In addition, 5 of the 157 differentially regulated genes regulated fetal sensory perception of the vision and olfactory systems. About 14% of the 157 genes including natriuretic peptide receptor A (*NPR1*) that regulates normal diuresis after birth were related to fetal physiology.

Cell-free fetal- and pregnancy-related DNA and RNA are more abundant in the amniotic fluid (AF) than in the maternal serum in the first and second trimesters when prenatal screening is predominantly performed [28,29]. As the fetal sample is uncontaminated by the maternal- and trophoblast-derived nucleic acids, cff nucleic acids present in AF are more representative of fetal gene expression [25]. To date, cultured amniocytes have been used to examine pure fetal nucleic acids in AF [30,31]. Amniocytes have been effectively used to identify gene expression patterns, but prolonged culture may result in genetic changes. As gene expression is context-dependent and is regulated by many factors, including the tissue of origin, environmental changes, developmental stage, nutrient availability, sex, and disease states, amniocyte culture may provide misleading results [25].

## 2. Amniotic Fluid

AF is a complex, dynamic environment containing growth factors, cytokines, and nutrients and plays an essential role in fetal development and protection during pregnancy by serving as a mechanical and functional barrier [32]. It can be broadly divided into two fractions, i.e., cells—termed “amniocytes”—and non-cellular liquid. AF is initially formed when maternal plasma passes through fetal membranes in the first trimester. At later stages of pregnancy, AF is enriched in excretory products from the fetal urine, respiratory system, gastroenteric system, umbilical cord, and the surface of the placenta, resulting in much more concentrated fetal material [33]. Amniocytes arise from all three germ layers of the embryo, ranging from unspecified progenitors to mature differentiated cells, including those of the renal, heart, lung, liver, and hematopoietic cell lineages [34]. During programmed cell death, amniocytes release cell-free transcripts. AF supernatant (AFS) represents the separated fraction without amniocytes after the centrifugation of AF. AFS contains a myriad of cell-free components, including the placenta-derived microparticles, proteins, cff DNA, and cff RNA [25]. In addition to the fetal-brain-specific transcripts, the AF transcriptome may also contain transcripts from fetal tissues that are not in immediate contact with the AF [28]. During the first half of gestation, diffusion between the fetus and the AF is bidirectional as the fetal skin is not keratinized. This permeable epidermal barrier may provide an exit for cff RNA originating in diverse tissues to contribute to the cff RNA found in the AF [35].

## 3. Cell-Free Fetal Nucleic Acids in Amniotic Fluid

AF contains abundant fetal cell-free RNA and DNA without maternal nucleic acid contamination. In 2001, the AF was found to be approximately 100–200-fold more enriched (per milliliter of AF) in fetal DNA compared with the maternal plasma using real-time quantitative polymerase chain reaction (PCR) [29]. Additionally, several researchers [36,37,38] have reported that there is no direct correlation in the amount of target-specific cff DNA between AF and maternal plasma, suggesting that the observed increase in the levels of maternal serum cff DNA is not the consequence of direct trafficking from the AF. This finding showed that the cff DNA in AF is physiologically distinct from the cff DNA in the maternal plasma [25]. Initially, real-time quantitative PCR for detecting *β-globin* expression and the *FCY (DYS1)* locus was used for identifying male DNA [29]. The methods for isolating cff DNA from AF have been refined to produce large quantities of cff DNA for subsequent analyses such as comparative genomic hybridization (CGH) microarrays. CGH microarrays have been successfully used to identify fetal sex and whole-chromosome gains or losses including trisomy 21 and monosomy X [39,40]. All 11 cff DNA samples from euploid male fetuses showed significantly increased hydridization-signal intensity for *SRY* and a decreased signal intensity for the X-chromosome compared with female DNA. On the other hand, cff DNA from six female fetuses had significantly decreased hybridization intensity for *SRY* and increased signals for the X-chromosome. In addition, three aneuploid cff DNA samples showed significantly increased hybridization signals on most of the chromosome 21 markers. On the contrary, cff DNA from a fetus with Turner syndrome (monosomy X) showed decreased signals for the X-chromosome. A recent study showed that cytogenetic technologies could elucidate whole-chromosome level changes in the fetus. Without time-consuming cell culture, advanced whole-genome sequencing and analysis of fetal genomes—utilizing cell-free DNA from AF—provided insight into nearly all small variants, copy number variants (CNVs), and structural variants (SVs) found in the parental genomes [41]. Although fetal DNA analysis can provide insights into the presence of fetal genetic material, fetal mRNA transcripts can provide additional valuable information about the gene expression patterns in fetal tissues. The dynamic nature of mRNA in AF has the potential to provide a better understanding of real-time fetal physiology and development [42].

The AFS transcriptome is a useful tool for providing a snapshot of fetal developmental processes at a distinct moment in pregnancy. The AF cell-free RNAs—contributed directly by the fetus and by apoptotic amniocytes—are altered by perturbations such as gestational age, sex, genetic syndromes, environmental exposure, and developmental stigmata [43]. Among these factors, the AF transcriptome seems to be most affected by gestational age with respect to the number of differentially expressed genes [44]. About 10% of the genes in AF (6194/64,071) showed expression changes with advancing gestational age from midtrimester to term pregnancy. These genes, which are about 11% of the coding and 6% of the non-coding genes, respectively, included 2776 increased and 3418 decreased genes (q-value < 0.05 and fold change >1.25). The AF transcriptome changes related to gene expression patterns have been associated with cell types found in the intrauterine environment and with the development of multiple organ systems [45,46,47,48]. Inducing genes according to advanced gestation were related to various signaling transduction pathways, surfactant and lipid homeostasis, cell immune response, and growth factors. In particular, genes involved in the vascular endothelial growth factor (VEGF) signaling pathway and surfactant function increased in AF with full term pregnancy. *SFTPA1, SFTPB, SFTPC,* and *SFTPD,* which are mRNA encoding surfactant proteins, and *LPCAT1, ABCA3, CTSH,* and *LYZ,* which are involved in lipid synthesis and precessing, and *FOXA2, NKX2-1,* and *HOXA5* that are related to lipid regulation increased in AF with advanced gestational time (weeks). These results can be indicative of the maturation of multiple organ systems besides fetal lung maturity [46,47].

Commercial kits are available for RNA extraction, amplification, fragmentation, and labeling. Isolated RNA following extraction is reverse-transcribed and amplified. After amplification, sense-stranded cDNA is fragmented, labeled, and hybridized to oligonucleotide microarrays. Discovery-driven research has enabled the extraction of sufficient cff RNA from AFS and subsequent microarray analyses [49,50,51]. In vivo evidence of the utility of using AF as a sample for the identification of a genetic disorder (twin–twin transfusion syndrome) in a living human fetus was first reported based on the significant upregulation of aquaporin 1(*AQP1*; a water transport gene) in AFS in a whole-transcriptome microarray. Additionally, gene expression varied with gestational age (GA) and sex [49]. Studies on single chromosome aneuploidy presented clinical distinct phenotypes arising not from simple gene dosage effects but from complex intragenomic transcription dysregulations in the second-trimester fetuses with Trisomy 21, Trisomy 18, and Turner Syndrome [50,52,53]. Non-genetic factors such as maternal obesity during pregnancy have also yielded convincing evidence of AFS whole-transcriptome differences regarding fetal neurodevelopmental and metabolic gene regulation between obese (BMI > 30) and lean women (BMI < 25) [54]. We have organized the studies on the AF cell-free transcriptome in chronological order (Figure 1). Therefore, based on the recent advances in this field, we reviewed the key scientific and clinical studies on the analysis of AF cell-free transcriptome in several common genetic and non-genetic diseases (Table 1).

## 4. Fetal Development and Placenta Cellular Dynamics

The AF cell-free transcriptome is modulated by physiological and pathological processes during pregnancy. In 2020, Tarca et al. [44] performed alternative splicing to detect the differentially expressed genes with advancing gestational age. About 17.5% (8566/48,820) of the genes showed a significant difference with gestational age from midtrimester to term gestation. The salivary gland-specific *MUC7*, lung-specific *SFTPD*, and stomach-specific *GKN1* genes were differentially expressed. This study also detected 64,071 genes in AF differentially expressed between the mid-trimester and term pregnancy. The most-enriched organs for genes, including the small intestine, placenta, and uterus and specific cell types (CD105+ endothelial cells and cardiac myocytes), are present during mid-trimester gestation. In contrast, this study found that the most overexpressed genes at term pregnancy were those responsible for the development of fetal organs such as tongue, trachea, salivary glands, tonsils, colon, and fetal lung. The gene expression for cardiac myocytes and the uterus slowly decreased but that for the trachea, salivary glands, and lungs increased throughout pregnancy.

There were both increased and decreased gene expressions that were associated with placenta development. Tsang et al. [55] conducted RNA-Seq analysis of the placenta and identified 13 cell types. Among them, the cytotrophoblast, syncytiotrophoblast, and monocyte expression increased gradually from 16 weeks of gestation to term pregnancy. On the other hand, the B cells, T cells, erythrocytes, Hofbauer cells, and vascular smooth muscle cells showed more complicated patterns. Based on single-cell genomics, the cytotrophoblast, which showed the average expression of *FAM3B, FOXO4,* and *MIR205HG,* was the most modulated and increased at term due to the contribution of the *FAM3B* gene. Therefore, AF mRNA profiles can be used to follow placental function as a revelation of the maternal-fetal interaction during pregnancy and showed physiological adaptations with advancing gestation in normal pregnancy.

## 5. Maternal Obesity

The rate of maternal obesity is increasing and is present in about one-third of pregnant women in the United States [56]. Maternal obesity may affect the neurodevelopmental and metabolic conditions of the fetuses. A previous study evaluated the whole-transcriptome differences in fetal gene expression between obese (BMI ≥ 30) women and normal (BMI < 25) women [54]. After matching the gestational age and fetal sex, 205 genes from fetuses derived from obese and normal pregnant women showed significant differences. Apolipoprotein D (*APOD*), a gene highly expressed in the central nervous system and integral to lipid regulation, was the most upregulated gene (9-fold) in the fetuses of obese women. Functional analysis using Ingenuity Pathway Analysis (IPA) revealed that genes associated with apoptosis were significantly downregulated, particularly in the cerebral cortex. Activation of the transcriptional regulator, the estrogen receptor (*ESR*)1/2, *FOS*, and *STAT3* was predicted in fetuses of obese women, suggesting a pro-estrogenic, pro-inflammatory intrauterine environment.

Eldow et al. [57] analyzed fetal gene expression at term using the umbilical cord blood to investigate the effects of maternal obesity on human development. Cord blood RNA was extracted and hybridized for gene expression assay after matching the gestational age and fetal sex. Seven hundred and one genes were found to be differentially regulated, and these were found to be related to neurodegeneration, indicating decreased survival of sensory neurons and decreased neurogenesis in the fetuses of obese women. Among 26 tissue-specific genes that were differentially regulated in the fetuses of obese women, six genes, i.e., *TSPAN7, NRGN, ENPP2, CHD5, CDC42*, and *C11orf95,* were highly expressed in the central nervous system. Additionally, upstream genes related to the inflammatory pathway were upregulated, but those associated with glucose metabolism, insulin receptor signaling, lipid metabolism, and response to oxidative stress were significantly downregulated.

## 6. Genetic Disorders

Genes commonly expressed in healthy euploid fetuses revealed that multiple organ systems were overrepresented in the AF transcriptome, including the hematological, musculoskeletal, and nervous systems [28]. Previous studies have demonstrated that expression differences between affected and normal fetuses are widespread in the AF transcriptomes of fetuses with Turner syndrome and trisomies 21 and 18 [50,52,53].

### 6.1. Trisomies 21 and 18

*Trisomies 21 and 18* are the most common genetic diseases and are caused by a partial or complete trisomy of chromosomes 21 and 18 [58].

Slonin et al. [50] investigated the difference in gene expression between normal pregnancy and trisomy 21 pregnancy during the second trimester. After matching the age and sex of seven fetuses, expression microarrays revealed that the 414 probe sets were significantly different between the euploid and trisomy 21 fetuses. The dysregulation of G-protein and ion transport signaling was noticed and it could cause abnormal development of neural and cardiac tissues. Among them, only five genes were located on chromosome 21. Another set of genes was located on chr21q22; these genes were significantly upregulated. To identify genes related to the trisomy 21 phenotype, the list of differentially expressed genes was investigated to reveal significantly disrupted cellular processes. This study suggested that reactive oxygen species disrupt the transport system and signal transduction, resulting in pathological symptoms associated with cardiac and neural tissue. Trisomy 18 has not been studied much compared to trisomy 21 because fetuses with trisomy 18 manifest anatomic abnormalities and are characterized by a high prenatal mortality rate (higher than 75%). Postnatally, the median survival time is 3–6 days and only 5–10% of the infants survive for one year [59]. Further, none of the regions on chromosome 18—which is associated with Edward’s syndrome—are known to be critical; this is different from the case for Down syndrome where region 21q22 is known to be critical for trisomy [60]. Koide et al. [50] compared the genome-wide transcriptome of euploid fetuses with those harboring trisomy 18 and found 251 genes to exhibit significant differential expression, of which only seven genes were located on chromosome 18. Among these, six genes were also differentially regulated in trisomy 21 compared to the control. In particular, Rho-associated kinase 1 (*ROCK1*)—located on chromosome 18—was significantly upregulated in trisomies 18 and 21. During early heart development, *ROCK1* plays an important role in the valvuloseptal formation and the regulation of endocardial cell differentiation and migration [61]. Adrenocorticotropic hormone (ACTH)—in conjunction with other factors, including fibroblast growth factor (FGF), epidermal growth factor (EGF), and insulin-like growth factor-1 (IGF-1)—is required for the normal morphological and functional development of fetuses. IPA revealed that these factors were downregulated in trisomy 18. Therefore, abnormal gene expression related to the adrenal network may contribute to fetal growth restriction, a well-known feature of trisomy 18 [52]. These findings are related to clinical features of trisomy 18, such as gross and microscopic hypoplasia of the adrenal cortical zone and valvular and endocardial cushion abnormalities.

### 6.2. Turner Syndrome

Turner syndrome is a common sex chromosome aneuploidy (prevalence, 1 in 2500 live births) [62] and in most cases results in miscarriage [53]. In 64% of the prenatal diagnostic cases and 47% of the postnatal diagnostic cases, monosomy X has been identified as the cause of Turner syndrome [63]. Clinical features and malformations, such as a webbed neck, short stature, coarctation of aorta, lymphedema, infertility, obesity, scoliosis, glucose intolerance, atherosclerosis, and juvenile rheumatoid arthritis are recognized for Turner syndrome, but no pathognomonic clinical characteristics have been identified [64]. Prenatal ultrasonography is used to detect fetal abnormalities, including increased nuchal translucency, fetal hydrops, and cystic hygroma [65]. AF could contribute to the understanding of the pathogenesis of Turner syndrome in the fetus. The residual AF supernatant is an abundant source of cell-free mRNA but is typically discarded [49].

Massingham et al. [53] investigated the cell-free mRNA transcriptome with monosomy X to reveal different gene expressions. There were only a few genes that exhibited differential expression that were common to monosomy X and trisomies 21 and 18. *XIST* is expressed commonly in all cells and is associated with X-inactivation. Decreased *XIST* expression was demonstrated in AF with Turner syndrome. The downregulation of another gene located on the X chromosome, i.e., *SHOX* is thought to be responsible for the short stature of women with Turner syndrome [66]. Excessive expression of immune system transcripts in the Turner syndrome may play an important role in the late onset of autoimmune disease observed in women with Turner syndrome [67,68]. These results implied that prevention of autoimmune disease could be a potential target for the in-utero therapy of Turner syndrome fetuses.

Turner syndrome is characterized by hypertension and coarctation of the aorta. A previous study identified upregulation of *NFATC3,* a gene whose expression is activated in response to intermittent hypoxia [69]. *NFATC3* is known to be involved in perivascular tissue and vascular remodeling via increasing the arterial wall thickness [70].

## 7. Pregnancy Complications

### 7.1. Preeclampsia

Preeclampsia (PE) is a specific obstetrical complication characterized by new-onset hypertension and proteinuria after 20 weeks of gestation. PE involves multiple organs, such as the kidney, liver, lung, and brain, and results in serious maternal and perinatal morbidity and mortality [71]. Various pathophysiologies have been implicated in PE, and one of the proposed models is a placental dysfunction caused by defective deep placentation [71].

In 2019, Jung et al. [72] investigated global gene expression and the AF-cell-free-transcriptome associated with the development of PE during the second trimester to identify biological pathways involved in the pathogenesis of PE. They showed that genes associated with the ribosome pathway and immune-related pathway were enriched in the AF- cell-free-transcriptome of PE. Enrichment of genes associated with the immune-associated pathway, including rheumatoid arthritis, graft-versus-host disease, and allograft rejection pathway in PE, indicated that disrupting immunomodulation could lead to PE. Additionally, genes associated with the ribosome pathway were enriched in the AF- cell-free-transcriptome of PE. Ribosomal RNAs (rRNAs) are involved in the ribosomal pathway, and the release of rRNAs into the extracellular fluid is related to apoptosis and necrosis [73]. Therefore, the abundance of rRNA in cell-free nucleic acids supports the increased placental apoptosis in PE [74]. This may reflect the importance of the immune-tolerance mechanism and stress response in the pathogenesis of PE.

### 7.2. Fetal Growth Restriction

Fetal growth restriction (FGR), i.e., the failure of fetal growth, is defined as estimated fetal weight below the 10th percentile for the gestational age. Small-for-gestational age (SGA) refers to weight below the 10th percentile for that specific gestational age without in utero growth failure [75]. In comparing the AFS transcriptome between FGR and normal fetuses, Cho et al. [76] found that the expression of insulin-like growth factor (IGF-2) was dramatically increased in FGR fetuses and this could be explained by the physiological change to compensate for energy distribution in a poor energy environment. In addition, the expression of low-density lipoprotein receptor (LDLR)-related protein 10 was significantly increased in FGR fetuses and it was found that this phenomenon could modulate fetal lung development via the canonical Wnt/β-catenin signaling pathway [77]. This could be related to accelerated lung maturation and pulmonary surfactant secretion in FGR fetuses.

### 7.3. Twin–Twin Transfusion Syndrome

The prevalence of twin–twin transfusion syndrome (TTTS) is about 5–10% of monochorionic diamniotic (MCDA) twin pregnancies [78]. TTTS occurs due to an imbalance of blood transfer between monochorionic twins across shared placental vascular anastomosis [79]. TTTS is known to affect fetal growth, the amount of amniotic fluid, neurodevelopmental function, and cardiovascular function and can be treated via amnioreduction of the recipient twin with prenatal laser ablation of the shared placental anastomosis [80]. Although most recipients recover normal cardiac function after successful laser ablation, they continue to be at a risk of developing structural anomalies and long-term abnormal cardiac function [81,82]. Some researchers have examined fetal gene expression to reveal differential gene expression in TTTS [47,49]. In 2005, *AQP1* was known to be the possible candidate gene responsible for causing TTTS; later, four genes, i.e., *AVPR1A FLT1, NRXN3,* and *NTRK3*—related to vasoconstriction, angiogenesis, synapse formation, cardiac, and neurological development, respectively—were confirmed to be related to TTTS using microarray. The dysregulated genes and associated pathways provide new molecular information regarding the development of nervous and cardiovascular systems and its relevance with respect to the long-term sequelae of TTTS.

## 8. Conclusions

AFS represents pure fetal samples that can provide gene expression information for living fetuses without amounting to an unpredictable risk to pregnancy. In fetal medicine, there are countless potential applications for AF cff RNA. AF cell-free transcriptomes provide valuable information on not only prenatal development of multiple fetal organs with advancing gestation but also placental dynamics during pregnancy. In addition, functional genomic analysis is a powerful tool for understanding both normal physiology and disease. Future studies are required to reveal the genes related to obstetrical complications. If candidate disease-related gene expression can be associated with the specific phenotypes, then the expression of these genes can assist us in understanding the pathophysiology of fetal diseases and provide insights into candidate biomarkers. Thus, AF containing AF cff RNA from multiple organs can be used to develop prenatal therapies in the future.

## Figures and Tables

**Figure 1 ijms-22-02612-f001:**
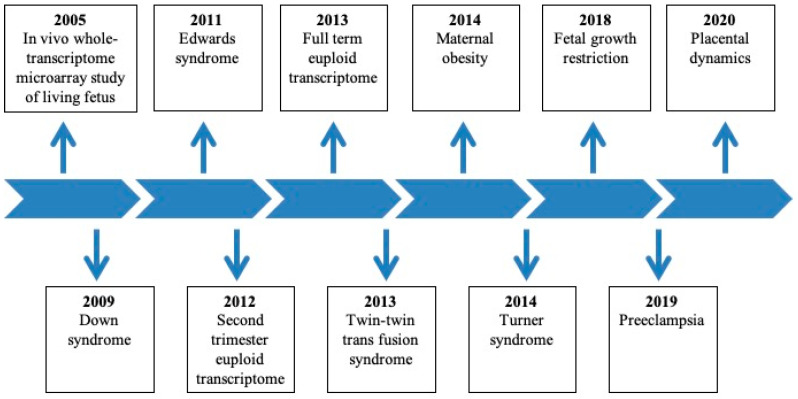
Amniotic fluid cell-free fetal RNA is a resource to study gene expression in normal development and fetal disease.

**Table 1 ijms-22-02612-t001:** Recent studies in fetal biology using cell-free fetal RNA in amniotic fluid.

Research	Sample Size	Condition Studied	Summary (Biomarkers Identified)
Larrabee et al. (2005)	5 (4 cases, 1 pooled control)	First in vivo whole-transcriptome microarray, fetal maturation	Four samples of AF cell-free RNA from 20–32 gestational weeks (GWs) with twin-to-twin transfusion syndrome (TTTS) were compared to an AF sample from pregnant women at 17–18 GWs. *AQP1* is well known as a water transport gene and it was up-regulated in TTTS fetuses. mRNA expression related to surfactant proteins, mucins, and keratins changed with advancing gestational age.
Slonim et al. (2009)	14	Down syndrome (DS, T21)	The dysregulation of G-protein and ion transport signaling was noticed and it could cause abnormal development of neural and cardiac tissues.
Koide et al. (2011).	11	Edwards syndrome (T18)	Rho-associated kinase 1 (*ROCK1*) was significantly upregulated in trisomies 18 and 21. During early heart development, *ROCK1* plays an important role in the valvuloseptal formation and the regulation of endocardial cell differentiation and migration
Massingham et al. (2011)	19	Fetal maturation	Standardized nanoarray PCR (SNAP) was performed to detect 21 transcriptional genes. MTOR and STAT2 gene expressions were higher in females than in males. *CASC3* and *ZNF264* genes were significantly increased according to gestational age while *ANXA5, GUSB*, and *PPIA* were decreased
Hui et al. (2012).	12	Normal fetal development	Twenty-three organ-specific transcripts such as the central nervous system, lung, tongue, and placenta were identified.
Hui et al. (2013)	16	Twin–twin transfusion syndrome (TTTS)	Four genes, *AVPR1A FLT1, NRXN3,* and *NTRK3,* related to vasoconstriction, angiogenesis, synapse formation, cardiac, and neurological development, respectively, were detected.
Edlow et al. (2014)	16	Maternal obesity	The dysregulation of *APOD* and activation of *ESR* 1/2, *FOS*, and *STAT3* affected the development of the central nervous system and lipid regulation.
Massingham et al. (2014)	10	Turner syndrome	Decreased *XIST* expression was observed with Turner syndrome. Genes that were related to perivascular tissue and vascular remodeling and the immune system were dysregulated.
Cho et al. (2018)	24 (15 cases, 9 controls)	Fetal growth restriction (FGR)	The expression of IGF-2 and LDLR-related protein 10 was increased in FGR fetuses and could be related to energy distribution and fetal lung maturation, respectively.
Jung et al. (2019)	82 (16 cases, 68 controls)	Preeclampsia (PE)	*RPS29, UBC*, and *IGF2* up-regulated genes were associated with the ribosome pathway. For functional annotation, the ribosomal and immune-related pathways were enriched in the AF

## Data Availability

Not available.

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
