# Peer review of "The Amniotic Fluid Cell-Free Transcriptome Provides Novel Information about Fetal Development and Placental Cellular Dynamics"

_ijms, 2021, doi:10.3390/ijms22052612_

Round 1

Reviewer 1 Report

Igms. 1128488:The amniotic fluid cell-free transcriptome provides novel information about fetal development and placental cellular dynamics

 General comments,

 The increasing number of scientific papers on AF cell-free transcriptomics indicates the importance of this approach in improving the comprehension of fetal development both in physiological and complicated pregnancy. However, although interesting, the title of  the present review  does not fit well with the whole manuscript: transcriptomics studies are only mentioned while they should be the most important part of the manuscript. I would suggest to reorganize the manuscript taking into account that a literature review should summarize the latest data on a given topic (i.e., AF transcriptome), uncover areas in which more research is needed, and also draw conclusions and suggestions for future studies.

Specific comments:

1.Introduction : The authors talked too much about DNA and mention RNA only once  at the end of the section; in addition they do not  indicate the purpose of the study. I would  suggest to insert  the section  2.Amniotic fluid  in the introduction

Section 3: This is the key section of the manuscript that needs a more detailed description (including the significance of the involved genes) of AF transcriptomics results

Section 4 and 5: can be unified in consideration of  the complex and tight interaction mother fetus

6.1 genetic disorders: Since there are few information on the syndromes  treated by the authors,  one section  may  be sufficient to summarize all data

And:

- a table summarizing literature data on AF transcriptome (including chromosomes, fetal sex, maternal health, genes and function)  will help readers to have a rapid idea of the state-of art

- please, write the full term of the acronyms

Author Response

Comments and Suggestions for Authors

Igms. 1128488:The amniotic fluid cell-free transcriptome provides novel information about fetal development and placental cellular dynamics

 General comments,

 The increasing number of scientific papers on AF cell-free transcriptomics indicates the importance of this approach in improving the comprehension of fetal development both in physiological and complicated pregnancy. However, although interesting, the title of the present review  does not fit well with the whole manuscript: transcriptomics studies are only mentioned while they should be the most important part of the manuscript. I would suggest to reorganize the manuscript taking into account that a literature review should summarize the latest data on a given topic (i.e., AF transcriptome), uncover areas in which more research is needed, and also draw conclusions and suggestions for future studies.

  • Thank you for your comment.

We added Table (Table 1) for summary of our manuscript and suggested more research areas for future studies in conclusion.

Specific comments:

Introduction : The authors talked too much about DNA and mention RNA only once  at the end of the section; in addition they do not  indicate the purpose of the study. I would suggest to insert  the section  2.Amniotic fluid  in the introduction

  • Thank you for your suggestion.

We added about fetal RNA in the introduction

Section 3: This is the key section of the manuscript that needs a more detailed description (including the significance of the involved genes) of AF transcriptomics results

  • Thank you for your comments

We wrote about significance of the involved genes in Section 3.

Section 4 and 5: can be unified in consideration of  the complex and tight interaction mother fetus

à Thank you for your suggestion.

Section 4 is about normal physiology while section 5 is about pathological process. We rewrite about section 4 and 5 more specifically instead of unifying two sections

6.1 genetic disorders: Since there are few information on the syndromes  treated by the authors,  one section  may  be sufficient to summarize all data

à Thank you for your comment!

We unified trisomies 21 and 18 and summarized genetic syndromes data with Table 1.

And:

- a table summarizing literature data on AF transcriptome (including chromosomes, fetal sex, maternal health, genes and function)  will help readers to have a rapid idea of the state-of art

à Thank you for your comment.

We made Table 1 for summary of our study.

- please, write the full term of the acronyms

à Thank you for your suggestion.

We added abbreviation data at the end of manuscript.

Comments and Suggestions for Authors

This is a well-written manuscript describing the potential of AF cell-free transcriptome in providing valuable information on prenatal development of multiple fetal organs as well as placental dynamics during pregnancy. The topic is covered broadly, the content are correct and update. This was very well-done.

  • Thank you for your comment

Reviewer 2 Report

This is a well-written manuscript describing the potential of AF cell-free transcriptome in providing valuable information on prenatal development of multiple fetal organs as well as placental dynamics during pregnancy. The topic is covered broadly, the content are correct and update. This was very well-done.

Author Response

Comments and Suggestions for Authors

This is a well-written manuscript describing the potential of AF cell-free transcriptome in providing valuable information on prenatal development of multiple fetal organs as well as placental dynamics during pregnancy. The topic is covered broadly, the content are correct and update. This was very well-done.

  • Thank you for your comment

Round 2

Reviewer 1 Report

none